# MULTI-AGENT ADVERSARIAL TIME SERIES FORECASTING

## ABSTRACT

Time series forecasting is critical across finance, energy, and healthcare, yet remains challenged by the complexity and non-stationarity of real-world data. Although deep learning has advanced performance, single-model architectures often struggle with temporal volatility and limited generalization. Multi-agent collaborative training offers a promising path forward by leveraging diverse model strengths; however, existing methods mostly rely on simple ensembles, lacking deeper structural interaction and probabilistic alignment. In this paper, we propose **M**ulti-**A**gent **A**dversarial **T**ime **S**eries **F**orecasting (MAA-TSF[1]), a framework that orchestrates heterogeneous generators and discriminators into a dynamic, competitive–cooperative system, akin to a multi-force formation adapting to evolving terrains. It integrates intra-group dynamic knowledge alignment and cross-group adversarial training to enhance joint distribution modeling and resilience to distribution shifts, while solving adversarial baseline instability. By evaluating nineteen real-world financial assets in six distinct market categories and six well-known datasets, we find that it consistently outperforms both the ERM and GAN under different time-specific backbones , achieving MAE reductions of $10\% - 70\%$, while delivering $5\% - 25\%$ gains in the accuracy of directional prediction across most datasets and models, verifying adversarial multi-agent coordination as a robust paradigm for complex time series.

## 1 INTRODUCTION

Multi-agent prediction, composed of distinct models, has emerged as a powerful paradigm for fitting complex real-world data distributions and generating realistic samples. Adversarial and collaborative dynamics within such an ensemble can be naturally framed as a joint distribution optimization problem. However, when multiple models are deployed on the same task, their performance remains bounded by architectural bottlenecks, heterogeneous input distributions, and training discrepancies, leading to limited generalization and robustness(Polikar, 2012; Dietterich, 2000). Therefore, viewing different modeling methods as the construction of an agent, how to find a system that can coordinate all agents to cooperatively optimize to achieve collective optimality while also improving individual generalization capabilities and performance on generating realistic data is a crucial problem.

In real-world data modeling, Time series forecasting plays a crucial role. In domains such as finance, energy, and healthcare, faces pronounced challenges due to real-world data properties—temporal distribution shifts(as shown in Figure 1), multi-scale fluctuations, and nonlinear dependencies(Shumway et al., 2000; Brockwell & Davis, 2002), conventional deep learning models, as well as classical ensemble methods such as Random Forests (Breiman, 2001) and Boosting, (Freund & Schapire, 1997), struggle to fully capture these complex temporal patterns. In these paradigms, competition is often prioritized over collaboration, which sacrifices substantial optimization benefits during training that stem from the randomness of various models. Furthermore, these methods are merely shallow ensembles that are independent of training processes and evaluated based on final performance.

---

[1]Code and data can be found at HERE.

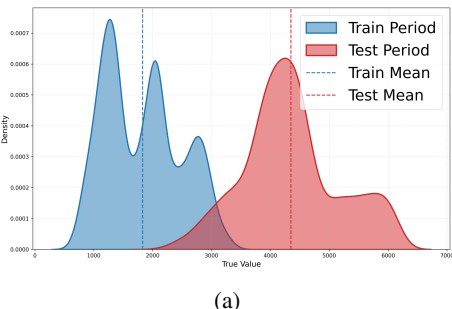 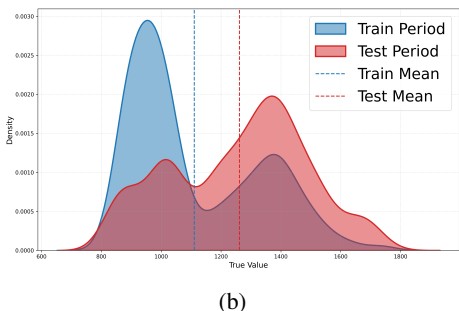

(a)                  (b)

Figure 1: Temporal Kernel Density Estimates (KDE) Visualization of Open Price Distribution Across Datasets: (a) S&P 500, (b) Soybean. The figure compares distributions between 01/2012-05/2020 (Train) and periods 05/2012-12/2024 (Test), highlighting significant temporal drift. Notable shifts in statistical properties (e.g., mean and variance) are observed across all asset classes.

Distributional shift across time severely limits the generalization capability of standalone or statically trained models(Fatima & Rahimi, 2024). Adversarial learning provides a dynamic pressure that encourages models to generalize by approximating the true data distribution under competition(Nam et al., 2024). However, GANs suffer from issues such as training instability, mode collapse, and difficulty in convergence (Arjovsky et al., 2017). Although methods like WGAN (Arjovsky et al., 2017) and WGAN-GP(Milne & Nachman, 2022) have partially stabilized the training protocol, they tackle the training-time mode collapse problem mainly from a regularization perspective, which, to some extent, limits the optimization search space. Therefore, finding an agent system paradigm capable of *Adversarial* and *Collaborative* is of paramount importance.

To address these challenges, we propose the **M**ulti-**A**gent **A**dversarial **T**ime **S**eries **F**orecasting framework (MAA-TSF). MAA-TSF integrates heterogeneous agents into a unified adversarial system, combining data-centric multi-task feature decomposition with model-centric adversarial & collaborative optimization. Through dynamic intra-group alignment and cross-group adversarial learning, MAA-TSF enhances both joint distribution modeling and agent-level generalization, offering a robust solution for forecasting complex, non-stationary time series. Our contribution can be summarized as follows:

**Present Works:** *1) MAA-TSF Framework*: We introduce MAA-TSF, a novel multi-agent adversarial framework for multi-task time series forecasting, addressing both modeling and optimization challenges posed by non-stationary data. *2) Dual-Level Learning Strategy*: We design a dual-level strategy that combines data-centric multi-task decomposition (e.g., trend, seasonal, noise separation) with model-centric dynamic collaboration (intra-group alignment with distillation and inter-group adversarial learning). *3) Extensive Empirical Validation*: We select 17 real-world financial asset of six categories datasets as an evaluation benchmark due to their inherent volatility and non-stationarity, which stress-test model robustness, demonstrating that MAA-TSF achieves superior performance compared to individual models and naive GAN. With the improved LSTM architecture yielding the most significant gains under high volatility conditions in the regression task and complex decision-oriented category task in the classification task.

## 2 PRELIMINARIES

### 2.1 PROBLEM FORMULATION

**Time Series Data.** Let $\mathcal{X} = \{x_1, x_2, \ldots, x_T\}$ denote a univariate time series, where $x_t \in \mathbb{R}$ represents the observation at time step $t$. For multivariate time series, we have $\mathcal{X} = \{\mathbf{x}_1, \mathbf{x}_2, \ldots, \mathbf{x}_T\}$, where $\mathbf{x}_t \in \mathbb{R}^d$ and $d$ is the dimension of the feature space.

**Time Series Forecasting Tasks.** Given a historical window of observations $\mathcal{W}_t = \{\mathbf{x}_{t-w}, \ldots, \mathbf{x}_{t-1}\}$ of length $w$, we consider two primary forecasting tasks:

*Regression Task.* The regression task aims to predict future values directly, formulated as $\hat{x}_{t:t+h} = f_R(\mathcal{W}_t; \theta_R)$ where $f_R$ is a forecasting model with parameters $\theta_R$, and $h \geq 1$ is the forecasting horizon. The objective is to minimize a regression loss function $\mathcal{L}_R$ between the predicted values and ground truth, typically based on a $p$-norm distance:

$$\mathcal{L}_R(\hat{x}_{t:t+h}, x_{t:t+h}) = \|x_{t:t+h} - \hat{x}_{t:t+h}\|_p, \tag{1}$$

where different choices of $p$ lead to various loss formulations commonly used in time series analysis.

*Decision-oriented Task.* Beyond direct value prediction, we also consider tasks that focus on deriving actionable insights from time series data. This can be formulated as: $\hat{y}_{t:t+h} = f_D(\mathcal{W}_t; \theta_D)$ where $f_D$ is a decision model with parameters $\theta_D$, and $\hat{y}_{t+h} \in \mathcal{Y}$ represents discrete states or actions derived from analyzing temporal patterns. The target $y_{t+h}$ can be obtained through various transformations of the original time series, capturing relevant state transitions or pattern changes.

The objective is to optimize a categorical loss function $\mathcal{L}_D$ that measures the discrepancy between predicted and actual decisions:

$$\mathcal{L}_D(\hat{y}_{t:t+h}, y_{t:t+h}) = \Phi(\hat{y}_{t:t+h}, y_{t:t+h}) \tag{2}$$

where $\Phi$ can take different forms depending on the specific decision/classification tasks, including categorical cross-entropy or specialized Value Functions($\mathcal{V}$) for structured prediction.

## 2.2 Multi-Agent Adversarial Framework

### 2.2.1 Competition

Our MAA-TSF framework consists of two groups of interacting agents designed to capture the complex joint distribution of time series tasks:

**Generative Group.** A set of $N$ generative models $\mathcal{G} = \{G_1, G_2, \ldots, G_N\}$, where each generator $G_i$ with parameters $\theta_{G_i}$ is capable of executing multiple tasks simultaneously: $\hat{\mathbf{O}}_{t:t+h_i}^{(i)} = G_i(\mathcal{W}_t^{(i)}; \theta_{G_i})$ where $\hat{\mathbf{O}}_{t:t+h}^{(i)} = \{(\hat{\mathbf{x}}_{t:t+h_i}^{(i)}, \hat{y}_{t:t+h_i}^{(i)}, \ldots), \ldots\}$ represents the complete set of outputs from generator $G_i$, potentially including regression values, classification decisions, and other task-specific predictions. Reversely, $\mathbf{O}_{t:t+h}^{(i)} = \{(\mathbf{x}_{t:t+h_i}^{(i)}, y_{t:t+h_i}^{(i)}, \ldots), \ldots\}$ represents ground truth. Each generator implicitly models a conditional distribution $p_{G_i}(\mathbf{O}_{t:t+h_i} | \mathcal{W}_t^{(i)})$ over the output space given the $G_i$-specific historical window.

**Discriminative Group.** A set of $M$ discriminator models $\mathcal{D} = \{D_1, D_2, \ldots, D_M\}$, where each discriminator $D_j$ with parameters $\theta_{D_j}$ evaluates the authenticity of the complete output set from a generator:

$$r^{(j)} = D_j(\mathcal{W}_t^{(j)}, \mathbf{O}_{t:t+h_j}; \theta_{D_j}) \tag{3}$$

where $r^{(j)} \in [0, 1]$ represents the probability that discriminator $D_j$ assigns to $\mathbf{O}_{t:t+h}$ or $\hat{\mathbf{O}}_{t:t+h}$ from any generator being a genuine continuation of the time series rather than a generated one. The discriminator assesses the joint consistency of all predictions with respect to the true data distribution.

### 2.2.2 Collaboration

**Distribution Perspective.** All tasks can be regarded as different marginal or conditional forms of the joint distribution $p(\mathbf{O}_{t:t+h} | \mathcal{W})$ (e.g. regression $p(\mathbf{x}_{t+h} | \mathcal{W})$ or classification $p(y_{t+h} | \mathcal{W})$). Generators learn this joint distribution, while discriminators evaluate its authenticity, explicitly modelling task inter-dependencies that are usually ignored when each task is trained in isolation.

**Multi-dimensional Distribution Alignment.** Knowledge sharing among heterogeneous models is driven by KL-Divergence:

$$\mathcal{L}_{\text{align}} = \mathbb{E}_{G_i, G_j} \text{KL}\big(p_{G_i}(\mathbf{O}_{t:t+h} | \mathcal{W}_t^{(i)}) \,\|\, p_{G_j}(\mathbf{O}_{t:t+h} | \mathcal{W}_t^{(j)})\big) \tag{4}$$

where the KL term jointly handles (i) *feature-space alignment* $(\mathbf{F}_i, \mathbf{F}_j)$ and (ii) *temporal-scale alignment* $(\mathcal{W}_t^{(i)}, \mathcal{W}_t^{(j)})$.

**Theorem 1.** *Entropy Equilibrium. Let $\bar{p} = \frac{1}{N}\sum_i p_{G_i}$ be the mixture distribution. Using the identity*

$$\frac{1}{N}\sum_i \mathrm{KL}\big(p_{G_i} \parallel \bar{p}\big) = H(\bar{p}) - \frac{1}{N}\sum_i H(p_{G_i}),\tag{5}$$

*minimising $\mathcal{L}_{align}$ is thus equivalent to minimising the system conditional entropy $H(p_{G_i})$ (up to the constant $H(\bar{p})$). Hence alignment promotes sharper, more confident predictions while still enforcing cross-model consistency.*

**Collective Minimax Optimisation.** Treating the whole generator set $\mathcal{G}$ and discriminator set $\mathcal{D}$ as two competing entities, we define

$$\min_{\mathcal{G}} \max_{\mathcal{D}} \mathcal{V}(\mathcal{G},\mathcal{D}) = \mathbb{E}_{\text{data}}\Big[\frac{1}{M}\sum_j \log D_j(\mathbf{O}_{G_i}) + \frac{1}{NM}\sum_{i,j} \log\big(1 - D_j(\hat{\mathbf{O}}_{G_i})\big)\Big]\tag{6}$$

or, equivalently, $\min_{\mathcal{G}} \mathbb{E}_{G_i}\big[\mathrm{KL}(p_{G_i}, p_{\text{data}})\big]$. This population-level game provides diversity and mutual teaching, yielding models that stay robust under distributional shifts.

## 3 METHODOLOGY

In this section, we embark on unveiling the intuition and details of our proposed paradigm. An overview of MAA-TSF (**M**ulti-**A**gent **A**dversarial **T**ime **S**eries **F**orecasting framework) is shown in Figure 2.

### 3.1 MULTI-AGENT ADVERSARIAL TRAINING

We formalize our multi-agent adversarial training procedure as a complex interaction between generator and discriminator collectives, each guided by distinct but interdependent objectives. Let $\mathcal{G} = \{G_1, G_2, \ldots, G_N\}$ represent our generator set and $\mathcal{D} = \{D_1, D_2, \ldots, D_N\}$ represent our discriminator set.

#### 3.1.1 TRAINING PROCEDURE FORMULATION

For each generator and discriminator, given historical windows $\{\mathcal{W}_t^{(i)}\}_{i=1}^N$ of potentially varying lengths $\{w_i\}_{i=1}^N$, the generators produce both a future length of $\{h_i\}_{i=1}^N$ value predictions and classification logits: $G_i(\mathcal{W}_t^{(i)}) = (\hat{\mathbf{x}}_{t:t+h_i}^{(i)}, \hat{\mathbf{p}}_{t:t+h_i}^{(i)})$, where $\hat{\mathbf{x}}_{t:t+h_i}^{(i)}$ represents the predicted values and $\hat{\mathbf{p}}_{t:t+h_i}^{(i)}$ represents the classification logits. The corresponding discrete classification prediction is:

$$\hat{y}_{t:t+h_i}^{(i)} = \arg\max_c \hat{\mathbf{p}}_{t:t+h_i,c}^{(i)}.\tag{7}$$

Each discriminator $D_j$ receives an input window paired with its continuation and produces a score indicating $Real$ or $Fake$: $D_j(\mathbf{X},\mathbf{Y}) \in [0,1]$ where $\mathbf{X}$ represents a sequence of latest regression value and $\mathbf{Y}$ represents the a sequence of latest categorical results(either real or generated). A score closer to 1 indicates the discriminator believes the continuation is $Real$, while a score closer to 0 indicates the discriminator believes the continuation is $Fake$, i.e. generated.

#### 3.1.2 CROSS-MODEL DISCRIMINATION WITH ADAPTIVE WINDOW

A key innovation in our approach is the cross-model evaluation framework. For each generator-discriminator pair $(G_i, D_j)$, we must align the window sizes while ensuring sufficient historical context for predictions.

We establish a fundamental constraint on window sizes:

$$\min_{i\in\{1,2,\ldots,N\}} w_i > h_{\max}\tag{8}$$

where $h_{\max}$ represents the maximum prediction horizon across all models. This constraint guarantees that even the model with the smallest window size has sufficient historical information to make reliable predictions.

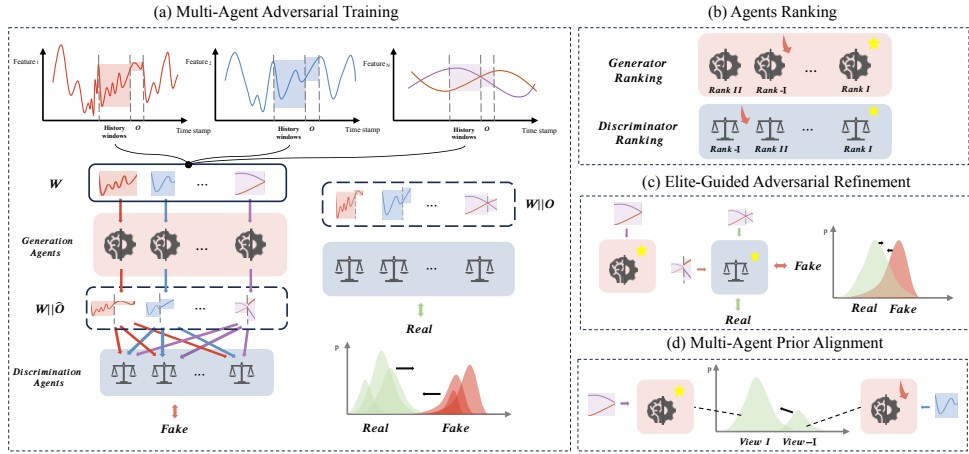

Figure 2: An Overall Architecture of MAA-TSF.

Within this constraint, we align inputs for cross-model evaluation as follows:

$$(\mathbf{X}, \mathbf{Y})_{i \to j} = \begin{cases} [(\mathbf{X}_i, \mathbf{Y}_i)_{(w_j - w_i):t}, G_i(\mathcal{W}_t^{(i)})], & \text{if } w_i < w_j \\ G_i(\mathcal{W}_t^{(i)}), & \text{if } w_i = w_j \\ G_i(\mathcal{W}_t^{(i)})_{(w_i - w_j):}, & \text{if } w_i > w_j \end{cases} \quad (9)$$

where $\mathbf{X}_i$ and $\mathbf{Y}_i$ are the real regression value and categorical label before the earliest forecasting according to the generator $G_i$.

This adaptive alignment ensures that regardless of the window size differences between generator $G_i$ and discriminator $D_j$, the input provided to $D_j$ maintains the expected temporal structure while preserving all relevant prediction information. The window size constraint further guarantees that pattern recognition is based on sufficiently rich historical contexts across all models in the system.

where $[\cdot, \cdot]$ denotes concatenation and temporal subscript indices denote slicing operations.

### 3.1.3 ADVERSARIAL TRAINING OBJECTIVE

**Discriminative.** When training discriminators, we optimize the following objective:

$$\mathcal{L}_{\mathcal{D}} = \sum_{j=1}^{N} \Big[ -\log D_j(\mathbf{X}_j, \mathbf{Y}_j) - \sum_{i=1}^{N} W_{ji} \log(1 - D_j((\mathbf{X}, \mathbf{Y})_{i \to j})) \Big] \quad (10)$$

where $W_{ji}$ represents an element from the weight matrix $\mathbf{W} \in \mathbb{R}^{N \times N}$ that modulates the influence of generator $G_i$ on discriminator $D_j$.

**Generative.** When training generators, we optimize a multi-objective function that balances prediction accuracy and adversarial deception:

$$\mathcal{L}_{\mathcal{G}} = \sum_{i=1}^{N} \Big[ \lambda_1 \|\hat{\mathbf{x}}_{t:t+h_i}^{(i)} - \mathbf{x}_{t:t+h_i}\|_2^2 + \lambda_2 \mathcal{L}_{\text{CE}}(\hat{\mathbf{p}}_{t:t+h_i}^{(i)}, y_{t:t+h_i}) - \lambda_3 \sum_{j=1}^{N} W_{ij} \log D_j((\mathbf{X}, \mathbf{Y})_{i \to j}) \Big] \quad (11)$$

where $\lambda_1$, $\lambda_2$, and $\lambda_3$ are hyperparameters controlling the relative importance of regression accuracy, classification accuracy, and adversarial deception, respectively. $\mathcal{L}_{\text{CE}}$ denotes the cross-entropy loss function.

**Dynamic Weight Matrix.** The weight matrix $\mathbf{W}$ plays a crucial role in modulating the interactions between generators and discriminators. It adapts during training based on the relative performance of different models:

$$W_{ij} = \frac{\exp(\beta \cdot \text{Perf}_{ij})}{\sum_{k=1}^{N} \exp(\beta \cdot \text{Perf}_{ik})} \quad (12)$$

where $\text{Perf}_{ij}$ measures the performance of generator $G_i$ against discriminator $D_j$ on the validation set, and $\beta$ is a temperature parameter controlling the sharpness of the weight distribution.

This formulation creates a dynamic adversarial ecosystem where generators and discriminators continuously adapt to each other's evolving strategies, driving the system toward increasingly accurate and robust predictions.

## 3.2 Elite-Guided Adversarial Refinement

While collective multi-agent training facilitates broad exploration of the model optimization space, focusing computational resources on the most promising models can significantly accelerate convergence and improve final performance. To this end, we introduce an elite-guided adversarial refinement process that selects and intensively retrains the best-performing generator-discriminator pair at regular intervals.

Specifically, after every $\kappa$ epoch of standard training (where $\kappa$ is a hyperparameter), we select the elite models based on their respective performance metrics.

$$i^* = \arg \min_{i \in \{1,2,\ldots,N\}} \mathcal{L}_{G_i}^{\text{val}}, \; j^* = \arg \min_{j \in \{1,2,\ldots,N\}} \mathcal{L}_{D_j}^{\text{train}} \tag{13}$$

where $\mathcal{L}_{G_i}^{\text{val}}$ represents the validation loss for generator $G_i$ and $\mathcal{L}_{D_j}^{\text{train}}$ represents the training loss for discriminator $D_j$.

The selected elite pair $(G_{i^*}, D_{j^*})$ undergoes an intensive adversarial training process for up to $\tau$ epochs. During this phase, the discriminator $D_{j^*}$ is updated to minimize:

$$\mathcal{L}_{D_{j^*}} = -\mathbb{E}_{\mathcal{W}^{(i^*)} \sim p_{\text{data}}}[\log D_{j^*}((\mathbf{X}_{j^*}, \mathbf{Y}_{j^*}))] - \mathbb{E}_{\mathcal{W}^{(i^*)} \sim p_{\text{data}}}[\log(1 - D_{j^*}((\mathbf{X}, \mathbf{Y})_{i^* \to j^*}))] \tag{14}$$

After updating the best discriminator, the generator $G_{i^*}$ is updated by minimizing:

$$\mathcal{L}_{G_{i^*}} = \lambda_1 \|\hat{\mathbf{x}}_{t:t+h_i}^{(i^*)} - \mathbf{x}_{t:t+h_i}\|_2^2 + \lambda_2 \mathcal{L}_{\text{CE}}(\hat{\mathbf{p}}_{t:t+h_i}^{(i^*)}, y_{t:t+h_i}) - \lambda_3 \mathbb{E}_{\mathcal{W}^{(i^*)} \sim p_{\text{data}}}[\log D_{j^*}((\mathbf{X}, \mathbf{Y})_{i^* \to j^*})] \tag{15}$$

## 3.3 Multi-Agent Prior Alignment with Knowledge Distillation

Although adversarial training establishes competitive dynamics among our agents, we incorporate knowledge distillation as a cooperative mechanism to efficiently propagate successful prediction strategies throughout the system. This approach creates a balance between competition and collaboration that improves the overall performance of the system.

**Performance-based Teacher-Student Pairing.** After each evaluation phase, models are ranked according to their performance on validation data, creating a performance hierarchy according to $\mathcal{L}_{G_i}^{\text{val}}$: $\mathcal{R} = \{i_1, i_2, \ldots, i_N\}$, where $i_1$ corresponds to the index of the best-performing generator and $i_N$ to the worst. We formulate a directed knowledge transfer from the highest performing generator (teacher) to the lowest performing one (student): $G_{i_1} \to G_{i_N}$.

**Multi-task Distillation.** Our knowledge distillation framework simultaneously transfers expertise between regression and classification tasks. For a given input batch, the teacher and student generate predictions:

$$(\hat{\mathbf{x}}_{t+1}^{(i_1)}, \hat{\mathbf{p}}_{t+1}^{(i_1)}) = G_{i_1}(\mathbf{X}_{i_1}), \;\; (\hat{\mathbf{x}}_{t+1}^{(i_N)}, \hat{\mathbf{p}}_{t+1}^{(i_N)}) = G_{i_N}(\mathbf{X}_{i_N}) \tag{16}$$

The distillation loss combines soft targets from the teacher and hard targets from the ground truth:

$$\mathcal{L}_{\text{distill}} = \alpha T^2 \cdot \text{KL}(p_{i_N}^T \parallel p_{i_1}^T) + (1 - \alpha) \cdot \left( \lambda_1 \mathcal{L}_{\text{CE}}(\hat{\mathbf{p}}_{t+1}^{(i_N)}, \mathbf{y}_{t+1}) + \lambda_2 \cdot \|\hat{\mathbf{x}}_{t+1}^{(i_N)} - \mathbf{x}_{t+1}\|_2^2 \right) \tag{17}$$

where $p_i^T = \text{Softmax}(\hat{\mathbf{p}}_{t+1}^{(i)}/T)$ represents the softened probability distribution using temperature $T$, $\alpha \in [0, 1]$ balances reliance on teacher knowledge versus ground truth, $T > 1$ is the temperature parameter that softens probability distributions, revealing richer informational content in teacher predictions.

A distinctive feature of our distillation framework is the asymmetric treatment of classification and regression tasks. The reason is that, for regression tasks, the ground truth values represent the optimal prediction target. Even the best teacher model inevitably contains prediction errors. Using the teacher's imperfect regression predictions as soft targets would introduce unnecessary bias into the student's learning trajectory when perfect targets (actual observations) are available.

# 4 EXPERIMENT

## 4.1 EXPERIMENT DESIGN

**Datasets.** Experiments are conducted on 19 real-world financial assets across six market categories and 6 well-known datasets. More data details are shown in AppendixB. All experiments are conducted on these 25 datasets, but twelve of them are shown in the main part, the others are shown in AppendixC. **Baselines.** To assess the performance of the proposed MAA-TSF[2] framework, we employ Empirical Risk Minimization (ERM) and a standard GAN (Goodfellow et al., 2014a) as baseline methods. For the standard GAN baseline, we evaluate various generator architectures, including Transformer(Vaswani et al., 2017), iTransformer(Liu et al., 2023), and PatchTST(Nie et al., 2022), paired with a CNN discriminator[3]. **Metrics.** Our evaluation framework assesses two distinct tasks: *Regression* (open step prediction) via Mean Absolute Error (MAE), *Directional Decision* (long/short/neutral position) via Accuracy. **Experiment Setting.** For 19 financial assets, we maintain consistent hyperparameters and backbone models, including a base learning rate of 2e-4, a hidden dimension of 512 for Transformer/iTransformer/PatchTST architectures, and window sizes of [5/10/15] (shared with their discriminators). For other 6 datasets the only difference is window sizes of [96/96/96] (shared with their discriminators). For MAA-TSF, we perform Elite-Guided Adversarial Refinement every 10 epochs and Multi-Agent Alignment every 30 epochs, with loss weights set to $\lambda_1=\lambda_2=\lambda_3=1/3$, and $\beta=1$ for adversarial dynamic weights. For alignment objectives, distillation parameters $\alpha=0.3$ (weight) and $T=2$ (temperature).

| Backbone | Baselines | Bitcoin | | Dow Jones | | Methanol | | Pulp | | Rubber | | Soybean | |
|---|---|---|---|---|---|---|---|---|---|---|---|---|---|
| | | Acc. ↑(%) | MAE ↓(E+01) | Acc. ↑(%) | MAE ↓(E+01) | Acc. ↑(%) | MAE ↓(E+01) | Acc. ↑(%) | MAE ↓(E+01) | Acc. ↑(%) | MAE ↓(E+01) | Acc. ↑(%) | MAE ↓(E+01) |
| Transformer | ERM | 51.51 | 88.93 | 62.32 | 170.86 | 53.77 | 5.67 | 54.07 | 1.63 | 54.15 | 43.83 | 59.29 | 2.11 |
| | GAN | 73.06 | 21.68 | 68.89 | **81.88** | 56.82 | 3.10 | 61.38 | **0.72** | 57.28 | 19.14 | 85.49 | 1.25 |
| | MAA | **84.80** | **15.98** | **74.05** | 82.78 | **60.94** | **2.45** | **63.19** | 0.82 | **58.56** | **13.80** | **86.81** | **1.16** |
| iTransformer | ERM | **90.92** | 74.92 | 85.50 | 16.74 | 74.67 | 2.92 | 74.71 | 0.20 | 74.73 | 12.13 | 88.25 | 1.06 |
| | GAN | 90.77 | **62.77** | 85.41 | 15.20 | **76.53** | 2.68 | **79.29** | **0.19** | **75.27** | **11.37** | 87.83 | 0.95 |
| | MAA | 90.88 | 65.46 | **85.55** | **14.41** | 76.11 | **2.38** | 75.42 | **0.19** | 74.70 | 11.67 | **88.50** | **0.93** |
| PatchTST | ERM | 55.57 | 140.23 | 70.25 | 30.94 | 58.26 | 4.19 | 48.21 | 0.32 | 49.95 | 19.60 | 53.69 | 1.81 |
| | GAN | 86.32 | 111.45 | 83.12 | 23.62 | **58.79** | **3.09** | 72.78 | **0.23** | 62.49 | 15.11 | 86.92 | **1.59** |
| | MAA | **87.41** | **107.00** | **83.76** | **23.21** | 58.26 | **3.09** | **73.63** | **0.23** | **63.69** | **14.90** | **88.19** | 1.59 |

| Backbone | Baselines | ETTh1 | | Exchange | | Weather | | Solar | | Traffic | | Electricity | |
|---|---|---|---|---|---|---|---|---|---|---|---|---|---|
| | | Acc. ↑(%) | MAE ↓(E+01) | Acc. ↑(%) | MAE ↓(E+01) | Acc. ↑(%) | MAE ↓(E+01) | Acc. ↑(%) | MAE ↓(E+01) | Acc. ↑(%) | MAE ↓(E+01) | Acc. ↑(%) | MAE ↓(E+01) |
| Transformer | ERM | 48.79 | 0.20 | 49.39 | 0.003 | 52.18 | 0.47 | 80.02 | 0.18 | 61.86 | 0.0005 | 53.75 | 38.07 |
| | GAN | 48.18 | 0.14 | 52.91 | 0.001 | 52.42 | 0.35 | 84.62 | 0.22 | 83.41 | 0.0003 | 53.51 | 37.06 |
| | MAA | **52.66** | **0.10** | **56.17** | **0.0005** | **56.66** | **0.19** | **89.47** | **0.11** | **89.23** | **0.0002** | **63.44** | **33.78** |
| iTransformer | ERM | 53.03 | 0.11 | 52.78 | 0.0005 | 54.96 | 0.24 | 88.01 | 0.16 | 85.96 | 0.0004 | 76.27 | 10.77 |
| | GAN | **53.51** | 0.11 | 52.54 | 0.0005 | 54.84 | 0.24 | 89.35 | 0.13 | 86.44 | 0.0004 | 76.51 | 11.30 |
| | MAA | 53.27 | **0.10** | **53.87** | 0.0005 | **55.57** | 0.20 | **89.59** | **0.10** | **90.31** | 0.0003 | **77.12** | **8.88** |
| PatchTST | ERM | 53.03 | 0.12 | 53.51 | 0.0007 | 52.91 | 0.26 | 88.62 | 0.25 | 84.47 | 0.0005 | 69.13 | 13.20 |
| | GAN | **54.36** | **0.10** | 54.36 | 0.0007 | 54.84 | 0.23 | 86.68 | 0.20 | 86.68 | 0.0005 | **71.07** | **10.47** |
| | MAA | 54.00 | 0.12 | **55.33** | 0.0007 | **55.93** | 0.22 | **88.50** | **0.11** | **88.98** | 0.0004 | 65.86 | 12.08 |

Table 1: Performance comparison of different on six financial assets and six well-known datasets. Bolded values highlight the best performance (lowest MAE or highest Decision Accuracy) among the baseline models, specifically compared within groups sharing the same backbone architecture, where applicable, for each metric and dataset.

## 4.2 ANALYSIS

From the tables, we observe that MAA-TSF predominantly outperforms both the ERM and GAN in both regression and decision tasks. Ablation Study is shown in AppendixD

---

[2]We drop the (-TSF) script in the table for succinct

[3]following the settings as standard GAN (Goodfellow et al., 2014a)

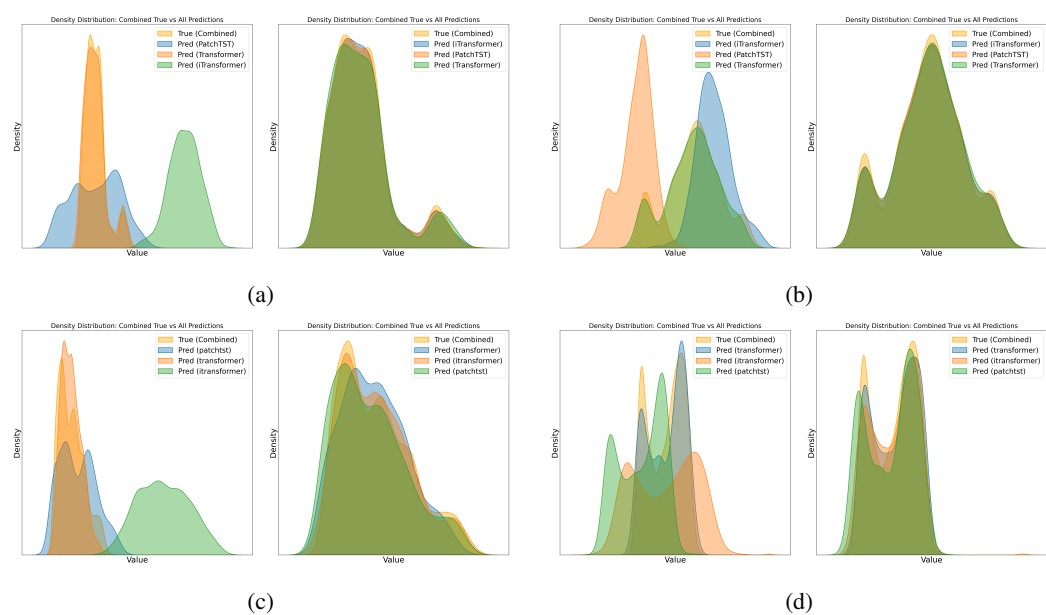

Figure 3: Comparative probability density distributions of forecasting results on dataset Rubber, Soybean, Weather and Traffic across four domains. **Left panels** show results from baseline models (Transformer/iTransformer/PatchTST) trained with stand-alone GAN, while **Right panels** present results from our MAA-TSF framework. Density estimates are computed using Gaussian kernel density estimation (KDE). The visualization compares true distributions (yellow fill) against predicted distributions (blue/orange/green lines for different models) on test slices. MAA-TSF demonstrates superior distribution alignment.

**Regression Prediction Performance.** Notably, MAA exhibits a substantial reduction in MAE, with performance improvements varying from 10% to 70% compared to ERM and 5% to 40% over GAN. For example, MAA achieves the lowest MAE across most datasets, including a reduction to 1.25 on Rubber, 0.95 on Pulp, 0.19 on iTransformer (Pulp), and 0.11 on Traffic, outperforming the best baseline by 5–30% depending on the dataset. The performance of Transformer in MAA improves significantly, with a reduction of prediction errors by about one-third to one-half across multiple datasets, including challenging ones like Bitcoin and Exchange. The Weather and Electricity datasets also demonstrate a noticeable improvement in MAA, especially in MAE, where it consistently delivers better performance than both ERM and GAN models.

**Directional Prediction Performance.** In terms of directional prediction accuracy (trend forecasting), MAA exceeds the best alternative by a margin of 5% to 25%. Specifically, MAA achieves the highest directional prediction accuracy in most datasets, reaching 90.77% on Pulp, 88.83% on Soybean, 85.51% on Dow Jones, 87.41% on PatchTST (Soybean), and 88.98% on Traffic, outperforming Transformer - ERM and iTransformer - GAN by 3% to 10% in these datasets. All three backbone models (Transformer, iTransformer, PatchTST) show improvements when integrated with MAA, but the greatest gains in directional accuracy are observed with Transformer. Furthermore, despite Transformer traditionally weaker performance in some prediction tasks, MAA enables PatchTST to surpass GAN and ERM in most datasets for directional accuracy, highlighting its robustness even on backbones that might not be as strong in certain scenarios.

**Volatility advantage.** Beyond overall improvements in regression and trend prediction, MAA-TSF also shows unique advantages under volatile market conditions (Figure 1), such as Bitcoin, Soybean, and the Dow Jones. Traditional single-model approaches and even vanilla GANs often struggle in these settings, frequently encountering issues like mode collapse or unstable convergence.

**Adversarial Robustness Generalization.** MAA-TSF delivers robust improvements in scenarios where adversarial learning tends to falter. Although GAN-based models offer expressiveness, they often suffer from mode collapse or unstable convergence, particularly in volatile markets. For ex-

ample, on the Natural Rubber dataset under the Transformer architecture, the MAE of the GAN model surges to 0.22, higher than the baseline ERM at 0.18, reflecting typical adversarial instability. In contrast, MAA-TSF reduces the error to just 0.11, highlighting its ability to stabilize training and correct adversarial drift. This advantage arises from its multi-agent design, which coordinates diverse generative paths and enforces regularized dynamics, leading to more reliable and accurate forecasts in challenging environments. We further show the distribution of ground truth and the prediction of different backbone models with Gaussian kernel density estimation (KDE) in Figure 3. Those trained with GAN perform severe mode collapse or overfitting on the train set, e.g., iTransformer on Rubber and PatchTST on Soybean, while our MAA-TSF paradigm shows a better alignment on prediction distribution, showing a better generalization capability.

**Rare regression cases.** While MAA-TSF excels under volatile conditions, its advantages diminish in smoother low-volatility datasets such as the ETTh1 and Traffic, both of which are none financial datasets. In such cases, the data tends to follow more predictable trends, and simpler architectures like GANs with Transformer backbones can slightly outperform MAA-TSF, typically by less than five percent. This contrast suggests that the additional complexity and inter-agent interactions in MAA-TSF are most valuable when the forecasting task demands flexibility and heterogeneous representations, but may offer only marginal improvements for more stable time series.

## 5 RELATED WORKS

While Generative Adversarial Networks (GANs) have potential for modeling complex time series distributions, basic architectures face mode collapse and training instability, and existing multi-component GANs in cross-domain adaptation, multi-task optimization, and multi-level feature modeling have limitations when applied to time series forecasting, such as limited capacity for long-range dependencies, gradient conflicts, and lack of dynamic adaptation for non-stationarity. Meanwhile, Multi-task Learning (MTL) and Multi-task Knowledge Distillation (MTKD) approaches, though promising in various domains, rely on static teacher–student architectures lacking adaptability to evolving time-series data and rarely incorporate adversarial mechanisms. Our proposed MAA-TSF framework addresses these issues by building a cluster of heterogeneous generators with dynamic inter-group adversarial mechanisms and intra-group knowledge distillation, and embedding a dynamic knowledge-distillation loop in a multi-agent system where models alternate between teacher and student roles, coupled with inter-agent adversarial training. Detailed related works are discussed in Appendix A.

## 6 CONCLUSIONS

This work presents the Multi-Agent Adversarial Time Series Forecasting (MAA-TSF) framework—a novel training paradigm that integrates heterogeneous generators, discriminators, and knowledge distillation loops into a cooperative multi-agent adversarial system. Unlike conventional single models or GAN-based paradigms, MAA-TSF addresses their limitations by treating forecasting as an adversarial and collaborative game: specialized agents jointly explore the solution space to alleviate local optima and mode collapse while stabilizing long-term predictions. Its adversarial dynamics enable continuous mutual refinement, and the integrated knowledge distillation aligns agents' learned priors via explicit knowledge transfer, thereby enhancing adaptability to volatile, complex time-series patterns. Empirically, MAA-TSF achieves significant performance gains in challenging scenarios but shows diminishing returns in some datasets when paired with highly capable backbones like Transformer, iTransformer, and PatchTST.

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

# A More Related Works

## A.1 Transformer-based Time Series Forecasting Model

Transformer-based models have become a promising way for time series tasks. Time series prediction (with Mean Squared Error, MSE, as the loss function) and time series classification (with Cross-Entropy, CE, as the loss function) are two main tasks in time series. Enterprise Risk Management (ERM) is a fundamental principle that aims to minimize the loss (or error) of a model on the available training dataset, serving as a core strategy to learn patterns from empirical data and lay the groundwork for the model's predictive performance. For ERM training in two time series tasks, we just minimize those two losses for each task.

In time series regression tasks, such as electricity price forecasting, traffic flow prediction, and energy load forecasting, Transformer-based models (Nie et al., 2022; Liu et al., 2023; Chen et al., 2024; Zhou et al., 2021; Wu et al., 2021; Wang et al., 2024b) have addressed critical challenges of long-term dependency modeling and non-stationarity, with MSE loss serving as the primary objective to minimize prediction errors.

In time series classification tasks, such as ECG arrhythmia detection, human activity recognition, Transformer-based models (Wang et al., 2024c; Feofanov et al., 2025; Chen et al., 2025; Wen et al., 2024; Wang et al., 2024a) leverage self-attention to capture discriminative temporal patterns, with Cross-Entropy loss optimizing class separation.

## A.2 Adversarial learning and Adversarial learning in time series

Adversarial learning is a paradigm in machine learning that leverages the conflict and interaction between two or more competing entities to drive the optimization of a target task. Adversarial learning has expanded to diverse domains, including computer vision, natural language processing. Goodfellow et al. (Goodfellow et al., 2014b) introduces GANs, which demonstrates that a generator and discriminator can be co-trained to generate realistic synthetic data. Early works focus on addressing GAN-specific challenges: Radford et al. (Radford et al., 2015) propose Deep Convolutional GANs (DCGANs), which stabilizes training by using strided convolutions and batch normalization, enabling high-quality image generation. Mirza extends GANs to conditional settings (cGANs) (Mirza & Osindero, 2014) , allowing control over generated data via auxiliary labels. Recent advances focus on addressing scalability and generalization challenges. Karras et al. (Karras et al., 2019) introduces StyleGAN, which uses a style-based generator to produce highly realistic, controllable images, setting a new standard for GAN-based generation. In robustness, Carmon et al. (Carmon et al., 2019) analyzes the generalization gap in adversarial training and proposed techniques to reduce it.

In time series, we can also use GANs for different tasks. (Li et al., 2019) proposes an unsupervised multivariate anomaly detection method based on Generative Adversarial Networks. It considers the entire variable set simultaneously to capture latent interactions among variables and fully utilizes the GAN-produced generator and discriminator. (Vuletić et al., 2024) introduces a novel economics-driven loss function for the generator to make GANs suitable for classification tasks in a supervised learning setting and it investigates the use of GANs for probabilistic forecasting of financial time series. (Chatterjee et al., 2025) proposes a modified Conditional Wasserstein GAN with a Gradient Penalty (CWGAN-GP) to generate synthetic time-series data matching the original data distribution, training model with the mix of the synthetic data and the original data to enhance model.

## A.3 Ensemble Methods and Cross Adversarial Learning

Ensemble learning enhances generalization and reduces prediction error by combining multiple models and leveraging model diversity, rooted in bias-variance theory (Zhou, 2025). In time series forecasting, early methods combined statistical models like ARMA and STL through weighted averaging (Bates & Granger, 1969). The advent of machine learning brought algorithms such as Random Forest (Bagging) (Breiman, 2001) and gradient boosting (Boosting, e.g., XGBoost, Light-GBM) (Chen & Guestrin, 2016; Ke et al., 2017), significantly improving the capture of non-linear patterns. However, these models often rely heavily on feature engineering and fixed structures.

In the deep learning era, ensemble strategies integrated with neural networks, leading to paradigms like Stacking (Wolpert, 1992) and AutoML-based approaches such as Google's AutoTS (Wang et al., 2022). Hybrid models like Facebook's Prophet (Taylor & Letham, 2018) also emerged, combining decomposition with statistical/machine learning methods. While these methods improved forecasting, they commonly rely on fixed weights or static combination strategies, lacking dynamic adaptability to data changes. Furthermore, they typically perform shallow fusion by merely combining prediction results, failing to fully leverage the structural complementarity of diverse models (e.g., CNNs vs. RNNs) or incorporate deep interaction mechanisms like knowledge distillation (Hinton et al., 2015), adversarial learning, or alignment human-level knowledge in Large Language Models(Wang et al., 2024d). This limitation in achieving deep synergy motivates the dynamic competition-cooperation mechanism of our proposed multi-agent adversarial framework, enabling heterogeneous models to learn complementary representations.

While Generative Adversarial Networks (GANs) show potential for modeling complex time series distributions, the basic generator-discriminator architecture suffers from mode collapse and training instability (Arjovsky et al., 2017). To address this, multi-generator/multi-discriminator (multi-G/multi-D) architectures have emerged, leveraging competition and cooperation to enhance model diversity.

Existing multi-component GANs have shown progress in other domains: *Cross-Domain Adaptation*: Architectures like $\Delta$-GAN (Burnell, 2018) and DualGAN (Yi et al., 2017) employ multi-G/multi-D systems for unsupervised domain mapping and distribution alignment. However, for complex time series, a single generator's capacity for long-range dependencies may be limited. *Multi-Task Optimization*: Triple-GAN (Li et al., 2017), MGAN(Hoang et al., 2018) and MGMDcGAN (Huang et al., 2020) use dedicated components for different objectives or data modalities. Applied to time series prediction, gradient conflicts among multiple components can lead to training instability, especially with non-stationary patterns. *Multi-Level Feature Modeling*: Models such as IDSEGAN (Phan et al., 2020) utilize multiple discriminators focusing on distinct feature levels. For time series forecasting, a single generator might struggle to capture multi-modal distributions (e.g., trend, seasonality, anomalies), and these methods lack dynamic adaptation for non-stationarity. Overall, applying existing cross-adversarial learning to time series forecasting faces key challenges: the conflict between long-range dependency modeling and adversarial training gradients, gradient conflicts among multiple components, and the absence of dynamic strategies for non-stationarity. Our proposed MAA-TSF framework tackles these by building a cluster of heterogeneous generators (e.g., combining RNNs and Transformers) and designing dynamic inter-group adversarial mechanisms, alongside intra-group knowledge distillation, to enable multi-model collaborative evolution.

## A.4 MULTI-TASK ALIGNMENT

Multi-task learning aims to improve generalization and capture inter-task relationships by sharing parameters or transferring knowledge across related tasks—e.g., trend, seasonality, and anomaly detection in time-series forecasting. Knowledge distillation (Hinton et al., 2015) offers a mechanism for cross-task knowledge transfer, mitigating optimization imbalance caused by task heterogeneity.

Existing Multi-task Knowledge Distillation (MTKD) research explores task-specific transfer (Yang et al., 2022; Li & Bilen, 2020), as well as the optimization of unified representations (Liu et al., 2019). Representative methods include (i) introducing alignment-driven objectives, such as quadruplet loss or logit calibration, to better match inter-task relations (Xu et al., 2023); and (ii) developing architecture-agnostic frameworks that perform joint, cross-task distillation to learn task-invariant representations (Liu et al., 2019; Formont et al.). Some studies further enhance distillation effectiveness by tailoring the loss function (Xing et al., 2022).

While these MTKD approaches show promise in various domains, they exhibit limitations for dynamic time-series data: most rely on static teacher–student architectures that lack adaptability to evolving data characteristics and rarely incorporate adversarial mechanisms to foster model diversity or robustness against non-stationarity and multi-modal distributions.

Our proposed MAA-TSF framework alleviates these issues by embedding a *dynamic* knowledge-distillation loop within a multi-agent system, allowing heterogeneous models to alternate flexibly between teacher and student roles based on real-time data. By coupling this loop with inter-agent

adversarial training, MAA-TSF achieves both adaptive alignment and enriched representation learning for complex time-series forecasting.

## B    DETAILED EXPERIMENTS SETTINGS

**Datasets** Experiments are conducted on 19 real-world financial assets across six market categories—**Stocks** (S&P 500, Dow Jones, SSE50, CSI300, U.S. Dollar Index), **Fixed-income** (U.S. 10Y Treasury, China 10Y Bond), **Commodities** (Lumber, Pulp, Methanol, Natural Rubber, Rebar), **Agricultural** (Soybean, Corn), **Energy** (Crude Oil, Shipping Europe Line), and **Crypto** (Bitcoin)-to evaluate the robustness and generalizability of MAA-TSF. Moreover, Experiments are also conducted 6 well-known datasets- ETTh1, Weather, Solar, Exchange, Traffic, Electricity- with each dataset only using the first 3071 rows. For model training and evaluation, we partition the data into three subsets using a fixed ratio of 7:1:2, corresponding to training, validation, and test sets, respectively. Each 19 real-world financial assets across six market is characterized by a set of time series data with daily time stamp from 01/2012 to 12/2024, which includes the following features:

• Price and volume: open, high, low, close price, and trading volume

• MA factors: 5-day, 15-day, and 30-day simple moving averages

• DIF: Difference between 12-day and 26-day exponential moving averages

• DEA: 9-day exponential moving average of DIF

• MACD(Appel, 1985): Moving Average Convergence Divergence, computed as DIF – DEA (typically based on 12, 26, and 9-day EMAs)

• ATR(Wilder, 1978): Average True Range, usually calculated over a 14-day window to reflect market volatility

• BOLL(Bollinger, 2002): Bollinger Bands, typically constructed using a 20-day moving average and ±2 standard deviations

• RSI(Wilder, 1978): Relative Strength Index, commonly computed over a 14-day period to capture momentum

• K, D, J(Lane, 1984): Stochastic Oscillator components, generally calculated using a 9-day period for K, with D as a 3-day simple moving average of K, and J derived accordingly to identify overbought/oversold signals

As shown in Table 2, ETTh1 dataset contain 7 variates collected from electric transformers from July 2016 to July 2018, which are recorded hourly and ETTm1/ETTm2 are recorded every 15 minutes. Electricity contains the electricity consumption of 321 customers from July 2016 to July 2019, recorded hourly. Solar collects production from 137 PV plants in Alabama, recorded every 10 minutes. Traffic contains road occupancy rates measured by 862 sensors on freeways in the San Francisco Bay Area from 2015 to 2016, recorded hourly. Weather collects 21 meteorological indicators, such as temperature and barometric pressure, for Germany in 2020, recorded every 10 minutes. ExchangeRate collects the daily exchange rates of 8 countries.

**Metrics** For financial assets, the regression task simply predicts next price. The decision task simplifies to predicting the next trading day's opening price movement direction (up/down/unchanged relative to the previous day), directly mapping to actionable trading strategies: long, short, or market avoidance. This accuracy metric complements MAE by quantifying the model's practical utility for trading decisions, where directional correctness matters more than precise value prediction. In the aspect of other six datasets, we use the same evaluation framework. For the regression task, we only evaluate the last feature column prediction results as the target feature. For the Directional Decision task, we don't consider the practical significance and just ensure the integrity of the experiment.

| Dataset | Domain | # Frequency | # Used Timestamps |
|---------|--------|-------------|-------------------|
| ETTh1 | Energy | 1 hour | 3071 |
| Solar | Energy | 10 mins | 3071 |
| Electricity | Energy | 10 mins | 3071 |
| Traffic | Traffic | 1 hour | 3071 |
| Weather | Environment | 10 mins | 3071 |
| Exchange | Economic | 1 day | 3071 |

Table 2: The statistics of six well-known datasets

| Backbone | Baselines | China 10Y Bond | | Corn | | CSI300 | | Lumber | | Oil | | Rebar | |
|----------|-----------|-------|------|-------|------|-------|------|-------|------|-------|------|-------|------|
| | | Acc. ↑(%) | MAE ↓(E+01) | Acc. ↑(%) | MAE ↓(E+01) | Acc. ↑(%) | MAE ↓(E+01) | Acc. ↑(%) | MAE ↓(E+01) | Acc. ↑(%) | MAE ↓(E+01) | Acc. ↑(%) | MAE ↓(E+01) |
| | ERM | 54.88 | 0.03 | 59.14 | 5.02 | 58.49 | 7.07 | 50.27 | 1.28 | 59.60 | 0.18 | 52.32 | 19.92 |
| Transformer | GAN | 56.32 | 0.03 | 79.68 | 1.57 | 64.97 | 5.17 | 54.59 | 0.64 | 84.97 | 0.14 | 54.15 | 6.77 |
| | MAA | 61.63 | 0.02 | 81.09 | 1.33 | 79.34 | 3.83 | 57.71 | 0.64 | 87.13 | 0.10 | 56.82 | 14.01 |
| | ERM | 70.16 | 0.03 | 85.33 | 1.30 | 83.26 | 3.13 | 79.44 | 0.65 | 87.93 | 0.10 | 77.33 | 5.00 |
| iTransformer | GAN | 70.16 | 0.03 | 84.13 | 1.26 | 83.26 | 3.08 | 80.56 | 0.70 | 87.83 | 0.09 | 77.87 | 4.55 |
| | MAA | 70.24 | 0.03 | 84.26 | 1.26 | 83.17 | 2.36 | 79.43 | 0.61 | 88.50 | 0.09 | 77.10 | 4.27 |
| | ERM | 46.34 | 0.03 | 64.59 | 2.09 | 67.21 | 5.23 | 57.29 | 0.90 | 61.50 | 0.19 | 46.56 | 7.41 |
| PatchTST | GAN | 49.27 | 0.03 | 79.67 | 1.70 | 79.13 | 3.98 | 59.43 | 0.86 | 78.59 | 0.16 | 57.69 | 5.57 |
| | MAA | 48.62 | 0.03 | 80.11 | 1.65 | 79.02 | 3.84 | 60.00 | 0.78 | 85.34 | 0.16 | 58.12 | 5.51 |

Table 3: Performance comparison of different models on more datasets. Bolded values highlight the best performance (lowest MAE or highest Decision Accuracy) among the baseline models, specifically compared within groups sharing the same backbone architecture, where applicable, for each metric and dataset.

| Backbone | Baselines | Shipping Index | | SP500 | | SSE50 | | US 10Y Treasury | | USD Index | |
|----------|-----------|-------|------|-------|------|-------|------|-------|------|-------|------|
| | | Acc. ↑(%) | MAE ↓(E+01) | Acc. ↑(%) | MAE ↓(E+01) | Acc. ↑(%) | MAE ↓(E+01) | Acc. ↑(%) | MAE ↓(E+01) | Acc. ↑(%) | MAE ↓(E+01) |
| | ERM | 58.82 | 16.86 | 58.56 | 16.67 | 58.70 | 6.19 | 58.75 | 0.02 | 60.33 | 0.07 |
| Transformer | GAN | 57.84 | 17.89 | 65.97 | 12.40 | 69.62 | 3.23 | 83.70 | 0.007 | 84.01 | 0.07 |
| | MAA | 58.70 | 14.01 | 65.08 | 16.10 | 77.81 | 2.52 | 87.91 | 0.005 | 88.98 | 0.04 |
| | ERM | 68.04 | 11.28 | 80.48 | 2.61 | 82.39 | 2.15 | 86.65 | 0.003 | 89.35 | 0.02 |
| iTransformer | GAN | 73.20 | 10.31 | 80.69 | 2.47 | 82.28 | 1.86 | 86.75 | 0.003 | 89.25 | 0.02 |
| | MAA | 69.57 | 9.62 | 80.38 | 2.39 | 81.86 | **1.81** | 87.60 | 0.003 | 89.19 | 0.02 |
| | ERM | 50.00 | 1.45 | 58.76 | 5.20 | 53.11 | 3.21 | 51.42 | 0.007 | 56.98 | 0.05 |
| PatchTST | GAN | 55.43 | 10.05 | 78.59 | 3.37 | 71.58 | 2.76 | 80.28 | 0.005 | 85.20 | 0.04 |
| | MAA | 51.09 | 10.02 | 78.90 | 3.30 | 77.92 | 2.63 | 83.43 | 0.005 | 86.67 | 0.04 |

Table 4: Performance comparison of different models on more datasets. Bolded values highlight the best performance (lowest MAE or highest Decision Accuracy) among the baseline models, specifically compared within groups sharing the same backbone architecture, where applicable, for each metric and dataset.

## C   MORE EXPERIMENTS RESULTS

Table 3 and 4 shows performance comparison of different models on other 11 financial assets. For most datasets, the MAA baseline consistently outperforms ERM and GAN in terms of MAE and Acc. Among backbones, iTransformer tends to deliver more robust performance across datasets, as seen in its relatively high Acc and low MAE for most entries. Overall, the MAA baseline demonstrates effectiveness in enhancing prediction accuracy and reducing error.

## D   ABLATION STUDY

**Best number of generator-discriminator groups.** To analyze the optimal number of generator - discriminator groups, we use Transformer as the backbone and use the oil dataset. The 6 - group setup achieves the lowest MAE of 1.2112. In terms of Accuracy (ACC), the 3 - group configu-

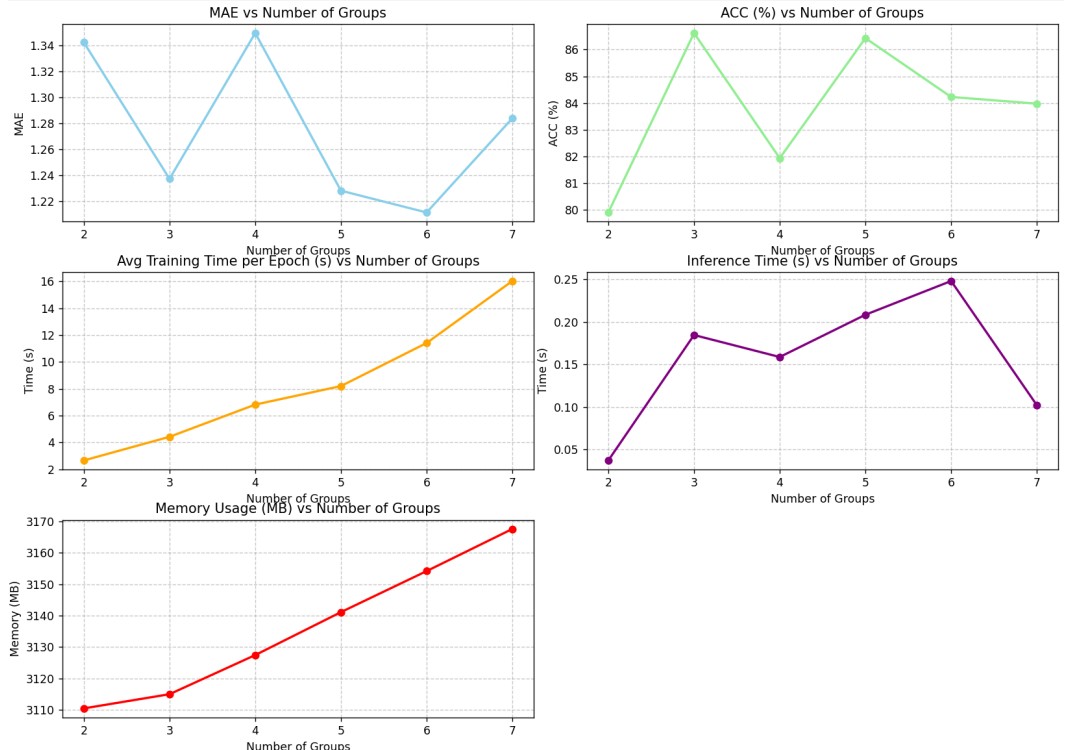

Figure 4: Performance of different numbers of generator-discriminator groups.

ration stands out with the highest ACC of 86.62%, which is crucial for direction prediction tasks. Considering computational efficiency, including training time per epoch, inference time, and memory usage, the 3 - group setup also performs well. Although the 6 - group has the best MAE, its training time (11.41s per epoch) and inference time (0.2480s) are relatively high, and memory usage (3154.19MB) is also considerable. Balancing all metrics—accuracy, error, and computational cost—the 3 - group configuration is the most optimal choice as it offers a good trade - off between performance and efficiency.

**Best generator-discriminator groups.** As shown in Table 5, to analyze how the combination of backbones influences the overall performance, we take the oil dataset and compare models with different backbone combinations (involving Transformer, iTransformer, and PatchTST). In terms of MAE, combinations with more powerful backbones, especially those including iTransformer, often achieve lower values. For example, in some setups, when iTransformer is part of the backbone combination, the MAE can be as low as around 0.9226, while combinations relying more on Transformer or PatchTST might have higher MAEs, such as Transformer - based combinations reaching 1.2475. Regarding ACC, combinations with stronger backbones also show better performance. When it comes to computational efficiency (training time per epoch, testing time, and memory usage), combinations with more powerful backbones, like those including iTransformer, also demonstrate advantages. Their training time per epoch can be around 3.70s, which is more efficient compared to combinations with PatchTST that take around 7.96s per epoch in some cases, and their memory usage and testing time are also relatively favorable. Although combinations with less powerful backbones like Transformer or PatchTST have their merits in specific metrics, combinations with more powerful backbones, leveraging their stronger capabilities, generally perform better across multiple metrics. Balancing all metrics—accuracy, error, and computational cost—combinations with more powerful backbones are more optimal choices as they offer better trade - offs between performance and efficiency, indicating that the more powerful the backbones in a combination, the better the overall effect.

**Different Backbones.** As shown in Table 6, 7, 8, we use relatively less advanced backbones—LSTM and GRU—and conduct tests on 17 financial assets. Compared with combinations of other backbones, the overall performance of the combinations of LSTM, GRU and Transformer

| Model | Training time per epoch | Testing time | Memory usage | MAE | ACC |
|---|---|---|---|---|---|
| Transformer | | | | 1.3283 | 0.7627 |
| Transformer | 5.33s | 0.1722s | 3116.54mb | 1.1445 | 0.8724 |
| Transformer | | | | 1.3404 | 0.8597 |
| iTransformer | | | | 0.8646 | 0.8776 |
| iTransformer | 3.13s | 0.0577s | 3088.97mb | 0.8690 | 0.8776 |
| iTransformer | | | | 0.8828 | 0.8755 |
| PatchTST | | | | 1.5828 | 0.8418 |
| PatchTST | 8.33s | 0.0890s | 3147.41mb | 1.5467 | 0.8576 |
| PatchTST | | | | 1.6369 | 0.8481 |
| Transformer | | | | 1.2475 | 0.8597 |
| iTransformer | 3.70s | 0.0579s | 3098.54mb | 0.9226 | 0.8703 |
| iTransformer | | | | 0.9697 | 0.8745 |
| Transformer | | | | 1.5828 | 0.8418 |
| PatchTST | 7.96s | 0.0885s | 3145.98mb | 1.5467 | 0.8576 |
| PatchTST | | | | 1.6369 | 0.8492 |
| iTransformer | | | | 0.8802 | 0.8734 |
| Transformer | 3.63s | 0.1363s | 3108.05mb | 1.3969 | 0.8713 |
| Transformer | | | | 1.4154 | 0.7268 |
| PatchTST | | | | 1.5828 | 0.8418 |
| Transformer | 7.81s | 0.0893s | 3150.05mb | 1.5467 | 0.8576 |
| Transformer | | | | 1.6370 | 0.8418 |
| Transformer | | | | 1.3387 | 0.8217 |
| iTransformer | 5.31s | 0.0764s | 3168.16mb | 0.9466 | 0.8692 |
| PatchTST | | | | 1.6255 | 0.8449 |

Table 5: Performance of different combinations of 3 generator-discriminator groups.

| Backbone | Baselines | S&P 500 Acc. ↑(%) | MAE ↓(E+01) | Shipping Europe Line Acc. ↑(%) | MAE ↓(E+01) | Bitcoin Acc. ↑(%) | MAE ↓(E+01) | CSI300 Acc. ↑(%) | MAE ↓(E+01) | Methanol Acc. ↑(%) | MAE ↓(E+01) | U.S. Dollar Index Acc. ↑(%) | MAE ↓(E+01) |
|---|---|---|---|---|---|---|---|---|---|---|---|---|---|
| GRU | ERM | 56.58 | 94.96 | 55.88 | 22.98 | 52.80 | 2690.28 | 53.51 | 21.18 | 50.33 | 5.56 | 47.35 | 0.19 |
| | GAN | 54.91 | 95.93 | 55.88 | 22.88 | 52.37 | 949.23 | 59.03 | 3.26 | 55.76 | 2.74 | 53.79 | 0.18 |
| | MAA | 55.59 | 80.84 | 53.26 | 20.62 | 85.89 | 492.56 | 63.72 | 3.55 | 53.83 | 2.57 | 54.04 | 0.20 |
| LSTM | ERM | 54.88 | 189.66 | 52.58 | 42.56 | 50.32 | 3510.49 | 53.48 | 49.04 | 50.53 | 12.36 | 52.61 | 0.72 |
| | GAN | 54.88 | 185.36 | 52.58 | 42.58 | 50.11 | 3507.22 | 53.48 | 47.38 | 50.53 | 12.15 | 52.61 | 0.09 |
| | MAA | 54.85 | 33.16 | 54.35 | 36.79 | 54.12 | 327.87 | 63.06 | 3.97 | 51.01 | 4.28 | 53.41 | 0.10 |
| Transformer | ERM | 55.06 | 36.21 | 52.17 | 17.02 | 54.05 | 451.23 | 56.28 | 5.07 | 49.40 | 4.52 | 53.31 | 0.19 |
| | GAN | 68.88 | 22.93 | 58.70 | 13.28 | 65.48 | 306.74 | 65.36 | 3.66 | 55.84 | 2.67 | 82.06 | 0.05 |
| | MAA | 72.68 | 14.16 | 58.70 | 12.99 | 83.50 | 252.34 | 78.80 | 3.77 | 58.79 | 2.64 | 83.32 | 0.05 |

Table 6: Performance comparison of different backbones on 19 finanicial assets.

| Backbone | Baselines | Soybean Acc. ↑(%) | MAE ↓(E+01) | Rebar Acc. ↑(%) | MAE ↓(E+01) | Dow Jones Acc. ↑(%) | MAE ↓(E+01) | SSE 50 Index Acc. ↑(%) | MAE ↓(E+01) | China 10Y Bond Acc. ↑(%) | MAE ↓(E+01) | Corn Acc. ↑(%) | MAE ↓(E+01) |
|---|---|---|---|---|---|---|---|---|---|---|---|---|---|
| GRU | ERM | 49.27 | 5.88 | 48.98 | 21.03 | 57.31 | 535.16 | 53.08 | 1.07 | 46.56 | 0.01 | 50.16 | 1.28 |
| | GAN | 55.64 | 2.50 | 48.98 | 20.70 | 67.54 | 68.09 | 65.19 | 2.15 | 46.56 | 0.05 | 56.22 | 3.92 |
| | MAA | 58.02 | 1.63 | 48.96 | 19.76 | 53.16 | 465.81 | 53.88 | 0.74 | 53.82 | 0.00 | 61.64 | 0.29 |
| LSTM | ERM | 50.26 | 22.83 | 51.41 | 57.94 | 53.10 | 1252.91 | 53.04 | 2.81 | 52.42 | 0.01 | 50.11 | 4.23 |
| | GAN | 50.26 | 22.82 | 50.33 | 58.99 | 53.10 | 1217.01 | 53.15 | 27.50 | 52.10 | 0.11 | 50.11 | 44.43 |
| | MAA | 84.39 | 1.18 | 51.04 | 8.45 | 53.16 | 158.21 | 59.89 | 0.29 | 53.01 | 0.00 | 50.05 | 0.41 |
| Transformer | ERM | 58.44 | 1.68 | 52.24 | 14.09 | 53.69 | 239.43 | 56.07 | 0.42 | 51.54 | 0.01 | 59.45 | 0.32 |
| | GAN | 71.84 | 1.28 | 57.47 | 9.28 | 81.75 | 88.49 | 63.17 | 2.80 | 56.10 | 0.03 | 79.23 | 1.53 |
| | MAA | 84.18 | 1.30 | 63.90 | 6.39 | 81.65 | 97.61 | 75.52 | 0.22 | 57.24 | 0.00 | 82.08 | 0.20 |

Table 7: Performance comparison of different backbones on 19 finanicial assets.

is poorer, which also shows that the more powerful the backbones in a combination, the better the overall effect.

| Backbone | Baselines | Natural Rubber | | US 10Y Treasury | | Lumber | | Crude Oil | | Pulp | |
|---|---|---|---|---|---|---|---|---|---|---|---|
| | | Acc. ↑(%) | MAE ↓(E+01) | Acc. ↑(%) | MAE ↓(E+01) | Acc. ↑(%) | MAE ↓(E+01) | Acc. ↑(%) | MAE ↓(E+01) | Acc. ↑(%) | MAE ↓(E+01) |
| GRU | ERM | 50.05 | 7.92 | **47.18** | **0.00** | 45.95 | 0.16 | 47.08 | 0.04 | **54.59** | 0.48 |
| | GAN | 50.27 | 18.63 | **47.18** | 0.04 | 45.95 | 1.60 | 52.82 | 0.42 | 53.03 | 2.42 |
| | MAA | **57.14** | **1.52** | 47.15 | **0.00** | **46.29** | **0.13** | **59.92** | **0.01** | 53.06 | **0.37** |
| LSTM | ERM | **50.11** | 11.78 | 47.32 | 0.01 | 45.56 | 0.41 | 53.20 | 0.09 | **53.10** | 1.20 |
| | GAN | 50.00 | 118.92 | **48.63** | 0.12 | 45.56 | 4.10 | 53.10 | 0.96 | 53.10 | 11.99 |
| | MAA | 50.05 | **2.47** | 47.15 | **0.00** | **46.29** | **0.10** | **63.71** | **0.01** | 53.06 | **0.15** |
| Transformer | ERM | 49.40 | 5.19 | 60.67 | **0.00** | 51.43 | 0.07 | 56.75 | 0.03 | 53.06 | 0.20 |
| | GAN | 58.02 | 13.61 | 72.15 | 0.02 | 60.57 | 0.62 | 81.01 | 0.14 | 62.03 | 1.19 |
| | MAA | **59.87** | **1.52** | **85.16** | **0.00** | **60.57** | **0.07** | **88.08** | **0.01** | **71.62** | **0.05** |

Table 8: Performance comparison of different backbones on 19 finanicial assets.

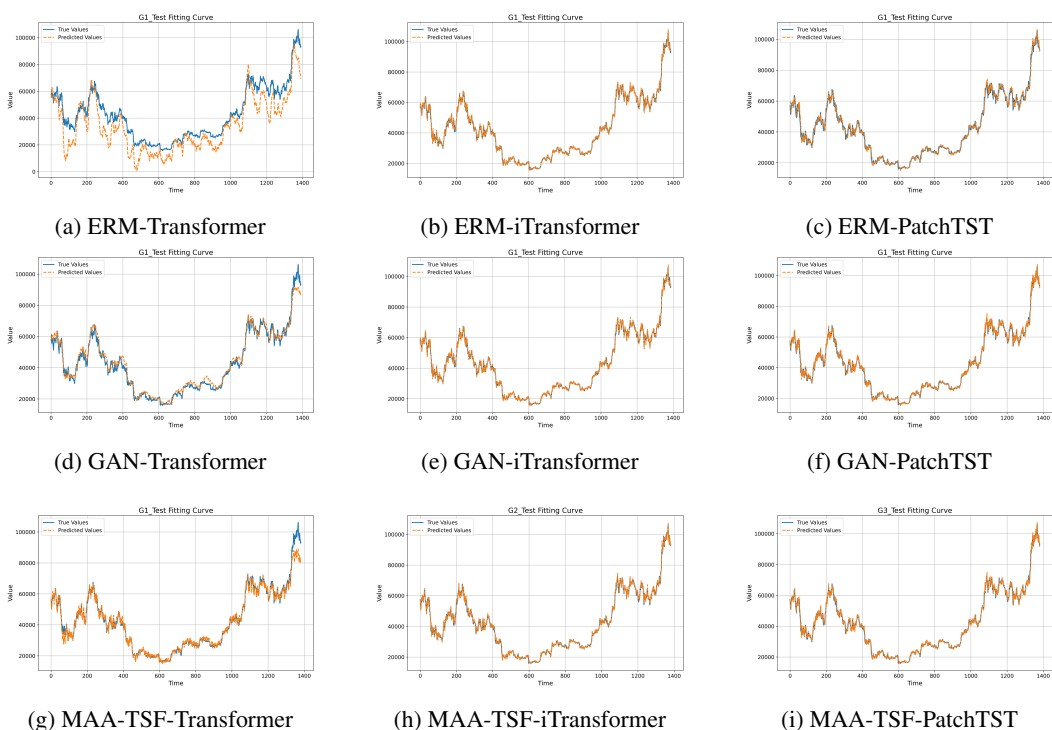

| (a) ERM-Transformer | (b) ERM-iTransformer | (c) ERM-PatchTST |
|---|---|---|
| (d) GAN-Transformer | (e) GAN-iTransformer | (f) GAN-PatchTST |
| (g) MAA-TSF-Transformer | (h) MAA-TSF-iTransformer | (i) MAA-TSF-PatchTST |

Figure 5: Fitting curves for Bitcoin dataset showing performance comparison across methods.

# E  USAGE OF GENERATIVE AI

We use several AI tools to assist in paper polishing and code implementation. Specifically, we use large language models such as ChatGPT and DeepSeek to polish our papers. All the content generated by them has been carefully reviewed by us and appropriately modified. During the development of the code, we also utilize Copilot and Cursors to optimize the code. All method designs and experimental analyses are independently completed by us and the use of artificial intelligence tools is strictly limited to auxiliary functions.

# F  VISUALIZATION

Each dataset evaluated in the main part is shown in a separate block with 3 rows (ERM, GAN, MAA-TSF) and 3 columns (Transformer, iTransformer, PatchTST), displaying the test set fitting curves. Rows from top to bottom: ERM, GAN, MAA. Columns from left to right: Transformer, iTransformer, PatchTST. Due to space limitations, we only present three set of visualizations. All visualizations are available in supplimentary materials.

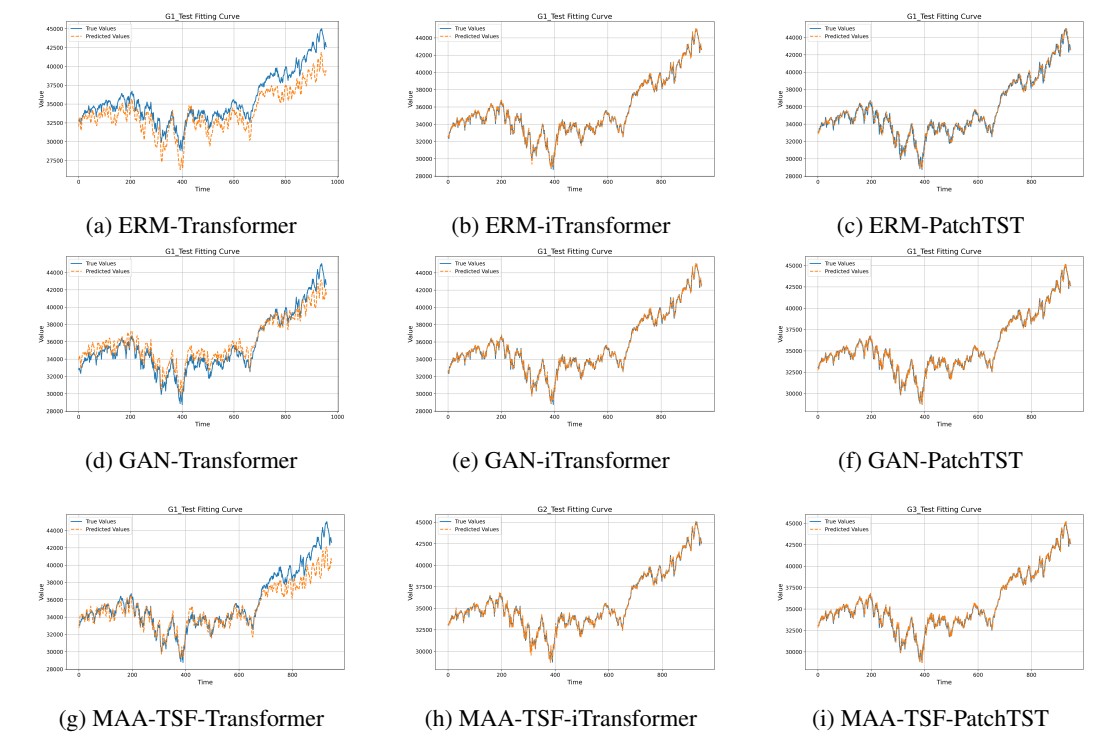

(a) ERM-Transformer (b) ERM-iTransformer (c) ERM-PatchTST

(d) GAN-Transformer (e) GAN-iTransformer (f) GAN-PatchTST

(g) MAA-TSF-Transformer (h) MAA-TSF-iTransformer (i) MAA-TSF-PatchTST

Figure 6: Fitting curves for Dow Jones dataset showing performance comparison across methods.

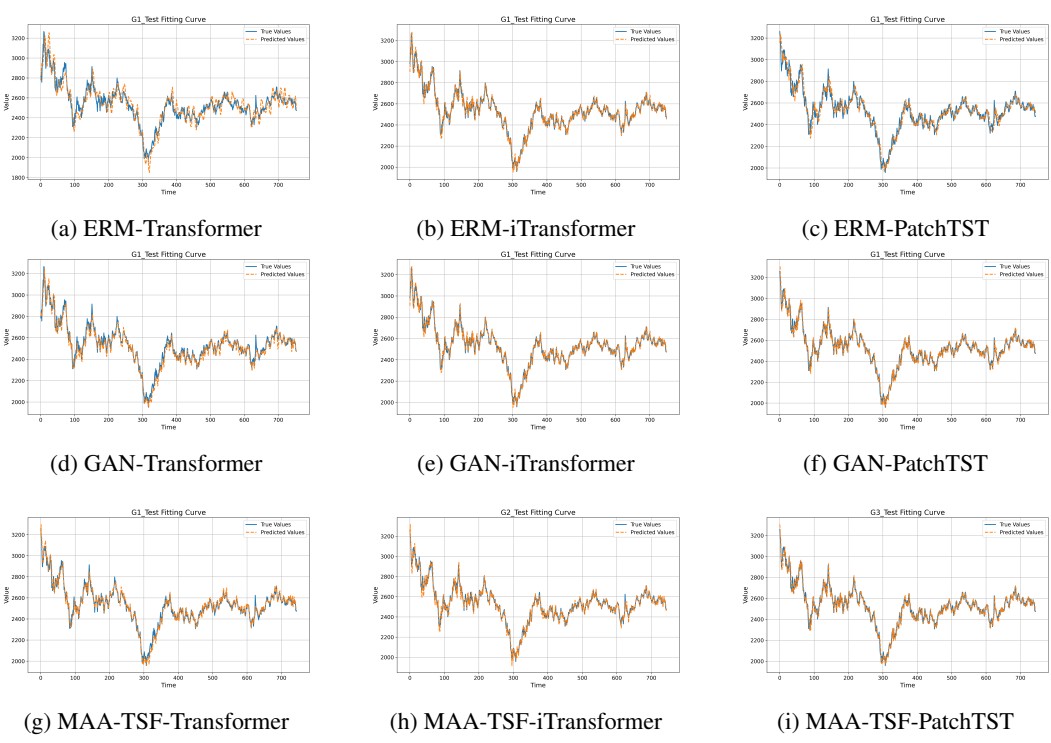

(a) ERM-Transformer (b) ERM-iTransformer (c) ERM-PatchTST

(d) GAN-Transformer (e) GAN-iTransformer (f) GAN-PatchTST

(g) MAA-TSF-Transformer (h) MAA-TSF-iTransformer (i) MAA-TSF-PatchTST

Figure 7: Fitting curves for Methanol dataset showing performance comparison across methods.

