# OpenReview forum: "Multi-Agent Adversarial Time Series Forecasting"
_ICLR.cc/2026/Conference — ICLR 2026 Conference Withdrawn Submission_

### Official Review · Reviewer_HCSC · 2025-11-01

**Soundness:** 2
**Presentation:** 3
**Contribution:** 2
**Rating:** 6
**Confidence:** 4

**Summary:**

This paper designs a dynamic, adversarial framework MAA-TSF to adaptively apply heterogeneous models for modeling complex time series. Leveraging intra-group alignment and cross-group adversarial training, the framework improves joint distribution modeling and robustness to distribution shift. Meanwhile, it is more stable than other adversarial baselines. Experiments span 19 financial assets across six market categories and six public time-series datasets and show consistent gains in MAE and directional accuracy, outperforming baselines.

**Strengths:**

- The research problem is well-motivated, targeting critical challenges in time series modeling.

- MAA-TSF is designed to model complex time series data by employing multiple models in an adversarial manner, aligning with recent trends in time series research.

- Evaluation spans diverse datasets, providing a comprehensive verification of the effectiveness of the proposed approach.

**Weaknesses:**

- This paper does not compare the proposed approach to strong baselines. In particular, simple ensemble methods are missing, which is specifically mentioned and compared to in the Introduction.

- The attached codebase is not accessible.

- [Minor] Figure 2 (a) is not readable enough for general audience unfamiliar with adversarial training, particularly the right-hand side part for the real distribution. A brief, self-contained caption may improve readability.

- [Minor] There are some small typos regarding the spacing, particularly before citations. No impact on my rating.

**Questions:**

- Are all experiments conducted with strictly chronological splits? Do the model performance and conclusions hold when using different train-test splits? Would its effectiveness be dependent on specific regimes?

- Can you compare MAA-TSF to the essential baselines, such as simple ensembles?

- Please interpret how and why MAA-TSF effectively addresses the distribution shifts between training and test data, using S&P 500 as an example? This is not clearly verified in the experiments.

- How do you determine the key hyperparameters, like $\lambda$s and $\kappa$? And how to interpret their impact?

I'm willing to raise my rating if my concerns are well-addressed.

---

### Official Review · Reviewer_5f99 · 2025-11-01

**Soundness:** 2
**Presentation:** 1
**Contribution:** 2
**Rating:** 2
**Confidence:** 5

**Summary:**

This submission aims to address the distribution shift problem in time series forecasting. To tackle this challenge, the manuscript proposes MAA-TSF, a framework designed to generate artificial training data. To evaluate the effectiveness of the proposed approach, fiveforecasting models combined with two data generation methods are tested across 25 datasets.

**Strengths:**

This submission presents a solid experimental evaluation, employing multiple baseline models and testing across more than 20 datasets to comprehensively assess the proposed method.

**Weaknesses:**

1. The submission is not well-written and lacks consistency in formatting. For example, the use of parentheses is inconsistent. In some instances, there is a space before “(”, while in others there is not. In addition, the formatting of figures and tables does not adhere to standard conventions; for instance, table titles should appear above the tables rather than below. Such issues detract from the overall readability and presentation quality of the paper.

2. The motivation of this submission is not sufficiently supported. The stated research goal is to address the issue of distribution shift, but the paper does not clearly explain why existing approaches, such as federated learning, reinforcement learning, or normalization techniques, among others, are inadequate for this purpose. Furthermore, it remains unclear why generating more data is a necessary or effective strategy for mitigating distribution shift. A clearer justification and theoretical or empirical evidence are needed to establish the significance of the proposed approach.

**Questions:**

Clarification of “Agent”: The paper should clearly define what an agent represents in the context of time series forecasting. It is currently unclear whether the agent refers to an individual forecasting model, a specific learning module, or a decision-making component within the proposed framework.

Definition of Joint Distribution Variables: The manuscript should explicitly specify the variables involved in the joint distribution mentioned in line 083. It is unclear what random variables or components this distribution models, for example, whether it involves the time series inputs, model predictions, latent representations, or other factors.

---

### Official Review · Reviewer_GYSc · 2025-11-03

**Soundness:** 2
**Presentation:** 1
**Contribution:** 2
**Rating:** 2
**Confidence:** 3

**Summary:**

The paper proposes Multi-Agent Adversarial Time Series Forecasting (MAA-TSF), a new framework to address the problem of the non-stationarity of real-world data in time series forecasting. MAA-TSF employs multiple generators and discriminators to learn the joint distribution and is robust to distribution shifts. MAA-TSF is experimented on 19 (12 in main paper) real-world datasets and shown to have better performance than competing baselines in terms of both point forecasting (MAE) and directional accuracy.

**Strengths:**

1. The idea of multi-agent adversarial training for time series forecasting is novel. The paper explores under-studied aspects such as distribution shifts and directional prediction, which are relevant and valuable problems in practical forecasting.

2. Experimental results on multiple datasets show promising performance.

**Weaknesses:**

1. The paper claims to address distributional shifts, but it is not evident how multi-agent adversarial training achieves this. No experiment explicitly evaluates performance under shifted distributions.

2. The roles of Elite-Guided Adversarial Refinement and Multi-Agent Prior Alignment with Knowledge Distillation are not clearly justified or empirically validated.

3. The paper does not directly compare with strong SOTA forecasting models. The competitiveness of results is unclear.

4. The paper is hard to read with convoluted explanations.

(i) Input to the discriminator is inconsistent between Eq. (3) and Eqs. (10–11).

(ii) Eq. (9) is confusing—why include part of the lookback window as generator output when it is already input to the discriminator?

(iii) Eq. (8) lacks justification for requiring lookback > forecast horizon.

(iv) Fig. 3 is unclear. Also, the font is too small in the figures.

(v) $\mathbf{F}_i$ (l. 160–161) is undefined.

(vi) Eq. (6) seems incorrect ($\sum_j \log(D_j(\mathbf{O}_{i})$?)

**Questions:**

See weaknesses

---

### Note · Authors · 2025-11-12

I have read and agree with the venue's withdrawal policy on behalf of myself and my co-authors.